# COVID-19 Vaccine Uptake Inequality Among Children: A Multidimensional Demographic Analysis

**DOI:** 10.3390/healthcare13091019

**Published:** 2025-04-29

**Authors:** Seyed M. Karimi, Mana Moghadami, Md Yasin Ali Parh, Shaminul H. Shakib, Hamid Zarei, Venetia Aranha, Sepideh Poursafargholi, Trey Allen, Bert B. Little, Demetra Antimisiaris, W. Paul McKinney, Yu-Ting Chen, Taylor Ingram, Angela Graham

**Affiliations:** 1Department of Health Management and Systems Sciences, School of Public Health and Information Sciences, University of Louisville, Louisville, KY 40202, USA; mana.moghadami@louisville.edu (M.M.); shaminul.shakib@louisville.edu (S.H.S.); hamid.zarei@louisville.edu (H.Z.); bert.little@louisville.edu (B.B.L.); demetra.antimisiaris@louisville.edu (D.A.); 2Department of Bioinformatics and Biostatistics, School of Public Health and Information Sciences, University of Louisville, Louisville, KY 40202, USA; mdyasinali.parh@louisville.edu (M.Y.A.P.); sepideh.poursafargholi@louisville.edu (S.P.); 3Department of Epidemiology and Population Health, School of Public Health and Information Sciences, University of Louisville, Louisville, KY 40202, USA; venetia.aranha@louisville.edu; 4Center for Health Equity, Louisville Metro Department of Health & Wellness, Louisville, KY 40202, USA; trey.allen@louisvilleky.gov (T.A.); yuting.chen@louisvilleky.gov (Y.-T.C.); taylor.ingram@louisvilleky.gov (T.I.); angela.graham@louisvilleky.gov (A.G.); 5Department of Health Promotion and Behavioral Sciences, School of Public Health and Information Sciences, University of Louisville, Louisville, KY 40202, USA; mckinney@louisville.edu

**Keywords:** COVID-19 vaccination, inequality, children, age, race, ethnicity, sex

## Abstract

**Background:** Although children can contract COVID-19, their typically lower immune reactivity appears to shield them from the intense hyperinflammatory response observed in other age groups, leading to milder disease symptoms. Nonetheless, children’s infection raises the possibility of unwanted transmission of the coronavirus to others, especially because most infected children are asymptomatic. **Objectives:** This study examines the uptake of the first and second doses of the COVID-19 vaccine among children by combinations of age, race, ethnicity, and sex. **Methods:** Data from the immunization registry were utilized to assess changes over time in COVID-19 vaccine uptake among children in Jefferson County, Kentucky’s most populous county. The analysis examined trends by age, race, ethnicity, and combinations of age-race, age-ethnicity, age-race-sex, and age-ethnicity-sex during the first six quarters of the COVID-19 vaccination rollout. **Results:** By May 2022, in 16–17-year-olds, the highest and lowest COVID-19 vaccination rates were observed among White and Black children (64.8% versus 41.2%, respectively, for dose two). The highest two-dose vaccination rate at ages 12–15 years was among Multiracial and White children (63.3% and 62.9%, respectively), the lowest among Black children (38.8%). The highest two-dose vaccination rate at ages 5–11 years was among children of Some Other Races, Asian, and White children (37.0%, 36.7%, and 35.5%, respectively), and the lowest among Black children (17.2%). **Conclusions:** Inequalities in COVID-19 vaccination based on race, ethnicity, and sex persisted throughout the study period. Efforts in vaccine distribution and promotional initiatives should focus on increasing vaccination rates among children from racial and ethnic minority groups and males.

## 1. Introduction

The pathogenesis of COVID-19—the biological mechanism underlying the disease—is marked by a complex interaction between the virus and the host immune response. Infection with COVID-19 can trigger an excessive immune reaction in some individuals, initiating a destructive cascade of inflammatory cytokines and chemokines that causes severe tissue damage. This hyperinflammatory response, rather than the direct effects of the virus itself, is often responsible for the severity of the disease and can lead to multi-organ failure and death in severe cases. Conversely, COVID-19 has been observed to manifest with milder symptoms and lower mortality rates in children compared to other age groups, a difference likely attributable to their distinct immunological system. Although children can be infected with COVID-19, their generally lower immune reactivity seems to protect them from the severe hyperinflammatory response seen in other age groups, resulting in less severe disease presentations [1,2].

Nonetheless, children can be infected with the COVID-19 virus. Since most infected children are asymptomatic, their infection raises the possibility of unwanted transmission of the coronavirus to others [3]. Furthermore, in May 2022, the Centers for Disease Control and Prevention (CDC) published a case definition for multisystem inflammatory syndrome in children (MIS-C) due to the increasing number of MIS-C cases caused by the COVID-19 virus [4]. MIS-C is a rare but serious condition that causes the spread of inflammation affecting various organs [5]. Another consequence of COVID-19 in children is its effect on children’s mental health and education [6,7].

Due to the consequences of COVID-19 on children and the fact that COVID-19 does not have a definitive treatment, vaccinating children could serve as a pivotal preventive strategy [8]. Most children were not authorized to receive a COVID-19 vaccine when the COVID-19 vaccination campaign started in the U.S. The first approved COVID-19 vaccine was authorized only for children who were 16 or older. It was the Pfizer COVID-19 vaccine that received an emergency-use authorization (EUA) from the United States (U.S.) Food and Drug Administration (FDA) on 11 December 2020 [9]. Moderna and Johnson & Johnson vaccines―which received EUAs from the FDA on 18 December 2020 [10] and 27 February 2020 [11], respectively―were authorized for individuals 18 and older. On 10 March 2021, the FDA expanded the Moderna vaccine’s EUA for 16 and 17-year-old children [12]. On 10 May 2021, and 29 October 2021, the FDA expanded the Pfizer vaccine’s EUA to 12–15-year-old and 5–11-year-old individuals, respectively [13,14]. Finally, on 8 December 2022, the FDA authorized updated bivalent COVID-19 vaccines for children as young as 6 months old [15,16].

Racial and ethnic minorities were affected more than others in the COVID-19 pandemic [17]. However, less data exist in the literature for children [18,19]. According to the CDC’s COVID-19 immunization survey, by August 2022, COVID-19 vaccination rates for at least one dose were highest among Asian children, with 63.4%, 89.2%, and 91.8% at ages 5–11, 12–15, and 16–17, respectively; Hispanic children had the second-highest rate at 34.5%, 63.9%, and 77.3%, respectively. For White children, the rates were 31.2%, 55.5%, and 64.6% at ages 5–11, 12–15, and 16–17, respectively; the corresponding rates for Black children were 29.8%, 56.6%, and 65.5% [19]. In addition to racial and ethnic inequality, sex differences in COVID-19 vaccine uptake have also been recorded, as 48.1% of females aged 5 to 17 were vaccinated, compared to 46.1% of males by August 2022 [19].

This study measures the COVID-19 vaccination trends by combinations of children’s age (namely, 5–11, 12–15, and 16–17 years), race/ethnicity (i.e., race or ethnicity), and biological sex (female and male). The combinations studied in the literature are sex-race-ethnicity, age-sex, and age-race-ethnicity, including in children’s studies [20,21]. To the authors’ knowledge, vaccination uptake in a combination of three or more demographic characteristics that include both age and sex (e.g., age-sex-race as in 5–11-year-old female Asian children, age-sex-ethnicity as in 5–11-year-old female Hispanic children, or age-sex-ethnicity-race as in 5–11-year-old female Hispanic Asian children) has not been studied.

The studied population consists of residents of Louisville/Jefferson County, Kentucky, USA. This urban county had a median household income of USD 67,849 in 2023 and an estimated population of 793,881 in 2024 [22]. Females slightly outnumber males in the county population (51.4% females versus 48.6% males) [22]. The county exhibits notable racial and ethnic diversity, with White residents comprising the majority (70.0%), followed by residents of Black (23.3%), Asian (3.5%), and Multiracial (3.0%), American Indian or Alaska Native (AI/AN) (0.2%), and Native Hawaiian or Other Pacific Islander (NH/OPI) (0.1%) races [22].

## 2. Materials and Methods

The Kentucky Immunization Registry (KYIR) was used to determine the number of COVID-19 vaccine doses administered during the first six quarters of the vaccination campaign, from December 2020 to May 2022, in Jefferson County, Kentucky. The KYIR data, provided by the Louisville Metro Department of Public Health and Wellness (LMPHW), include basic demographic details of vaccine recipients (namely, age in single years, biological sex, race, and ethnicity), the ZIP code of residence, vaccination dates and locations, and the type of vaccine. Race and ethnicity in the KYIR data are reported in categories used by the U.S. Census Bureau: seven races (AI/AN, Asian, Black, Multiracial―i.e., two or more races―NH/OPI, Some Other Races, and White) and two ethnicities (Hispanic and non-Hispanic).

The cumulative number of COVID-19 vaccinated children for each age-race group (e.g., 5–11-year-old Asian children or 5–11-year-old Black children), age-ethnicity group (e.g., 12–15-year-old Hispanic children or 12–15-year-old non-Hispanic children), age-race-sex group (e.g., 16–17-year-old Black female children or 16–17-year-old Black male children), and age-ethnicity-sex group (e.g., 5–11-year-old Hispanic female children and 5–11-year-old non-Hispanic male children) was calculated. Ethnicity-race groups (as in Hispanic Black, non-Hispanic Black, Hispanic White, non-Hispanic White, and so on) were not formed, as the number of children in all ethnicity-race groups, except for Hispanic White, was tiny to analyze.

The cumulative number of vaccine recipients in a demographic group was divided by the estimated population of the group to calculate the COVID-19 vaccination rate in a demographic group. For example, to calculate the two-dose COVID-19 vaccination rate among 5–11-year-old Black female children on a particular date, the cumulative number of 5–11-year-old Black female children who received at least two doses of one of the COVID-19 vaccines by that date was divided by the estimated population of 5–11-year-old Black female children in the county. We used Census 2010 and 2020 single-year age population data by ZIP code, sex, race, and ethnicity to estimate demographic groups’ population at county and ZIP code levels.

A child’s age, race, and ethnicity were determined based on what was reported when receiving the first dose. The choice of using the reported demographics at the first dose was made upon observing changes in the reported race over subsequent doses. The changes were usually from White or Black to Multiracial. Other scenarios examined were choosing the reported race at the last dose or the most frequently reported race over doses. The first led to the calculation of vaccination rates that were notably above 100% for some demographic groups, but the results from the latter were similar to when the first dose race was used and seemed realistic due to the county’s population’s racial mix according to the U.S. 2020 census. Therefore, the race reported at the first dose was used for simplicity and consistency. If a child’s demographic information was missing at the first dose, it was imputed by the relevant information from later doses. Two reported races (namely, American Indian or Alaska Native and Native Hawaiian or Other Pacific Islander) were excluded from the analyses due to their very small population in the county. Those with missing ethnicity were counted as non-Hispanic since they constitute approximately 92% of the county’s population. More details on the data refinement and organization are elsewhere [20].

Statistical proportion tests were conducted to compare vaccination rates between demographic groups within and across age groups at the final time point of the study, 31 May 2022. Specifically, the statistical significance of racial and ethnic differences in vaccination rates within age groups was tested by dose. Moreover, the statistical significance of the difference in doses one and two vaccination rates within age groups and the overall vaccination rate differences across age groups by dose was tested.

## 3. Results

The analyses included 13,120, 25,880, and 25,213 children from 16–17, 12–15, and 5–11 age groups, respectively, at the end of the study period, 31 May 2022. The distribution of vaccinated children by age, sex, race, and ethnicity is presented in Appendix A. The trends in the cumulative number of vaccinated individuals by race and ethnicity for each age group and dose are presented in Appendix A.

### 3.1. Inequalities in Uptake by Children Age-Race Groups

#### 3.1.1. 16–17-Year-Olds

The overall and race-specific COVID-19 vaccination rates among 16–17-year-old Jefferson County children increased rapidly from March 2021, when the Moderna vaccine received an EUA from the FDA for the age group, until October 2021. The rates continued to increase but at a slower pace from October 2021 to May 2022 (Figure 1a,b). By 31 May 2022, the overall dose-one and dose-two COVID-19 vaccination rates in this age group were 66.4% and 58.0%, respectively (Table 1).

During the study period, 16–17-year-old White residents consistently had the highest vaccination rate among all races, exceeding the overall rate by 7.3% for dose one and 6.8% for dose two at the end of the period (Figure 1a,b and Table 1). Asian 16–17-year-olds had the second highest dose-one and dose-two vaccination rates until October 2021 and March 2022, respectively, but were surpassed by Some Other Races thereafter. Children 16–17 years of both Asian race and Some Other Races had a higher than average dose-one and dose-two vaccination rate on 31 May 2022: 67.2% and 58.6% for Asian race and 71.7% and 60.0% for Some Other Races.

The trends in one-dose and two-dose vaccination rates among Multiracial 16–17-year-old children were similar to the overall trends such that they were, respectively, 1.6% below and 0.8% above the overall country-level rates at the age group on 31 May 2022.

Black 16–17-year-old residents consistently had the lowest COVID-19 vaccination rates among all races. Their dose-one vaccination rate was 48.7% on 31 May 2022, 17.7% below the overall vaccination rate (Figure 1a and Table 1); 41.2% for dose two, 16.8% lower than the overall rate (Figure 1b and Table 1).

#### 3.1.2. 12–15-Year-Olds

The overall and race-specific COVID-19 vaccination rates among 12–15-year-old Jefferson County children increased at an accelerating pace from May 2021, when Pfizer’s vaccine received EUA from the FDA for 12–15-year-old children, until October 2021 [23]. The rates continued to increase from October 2021 to May 2022, but at a much slower pace than before (Figure 1c,d). By 31 May 2022, the overall dose-one and dose-two COVID-19 vaccination rates in this age group were 63.9% and 56.1%, respectively (Table 1).

During the study period, 12 November 2020–31 January 2022, 12–15-year-old White residents consistently had the highest vaccination rate among all races: it reached 70.5% and 62.9% for doses one and two, respectively, on 31 May 2022 (Figure 1c,d and Table 1). However, by the last months of the study period, the dose-one rate among Multiracial children and children of Some Other Races and the dose-two rate among Multiracial children reached the rate in White children.

The trends in COVID-19 vaccination rates among 12–15-year-old Asian children were similar to the overall trend, although they were the second highest rates from May to August 2021 (Figure 1c,d).

Further, 12–15-year-old Black children consistently had the lowest COVID-19 vaccination rates among all races. Their dose-one vaccination rate was 45.9% on 31 May 2022, 18% below the overall vaccination rate (Figure 1c and Table 1); 38.8% for dose two, 17.3% lower than the overall rate (Figure 1d and Table 1).

#### 3.1.3. 5–11-Year-Olds

The overall and race-specific COVID-19 vaccination rates among 5–11-year-old Jefferson County children increased rapidly from November 2021, right after the Pfizer vaccine received an EUA from the FDA for the age group, until February 2022. The growth rate in vaccinating 5–11-year-old children slowed down thereafter. The overall dose-one and dose-two COVID-19 vaccination rates in this age group were 36.7% and 30.6%, respectively (Table 1).

In the first two months of COVID-19 vaccination among 5–11-year-olds, White children had the highest dose-one vaccination rate among all races (Figure 1e,f). Children of Some Other Race and Asian race surpassed them in January and February 2022, respectively, reaching the rates of 50.9% and 43.4% on 31 May 2022, versus 41.4% among White children (Table 1). Asian 5–11-year-olds had almost the same vaccination rates as White children. Children 5–11 years of Multi Races had a higher-than-average dose-one and dose-two vaccination rate on 31 May 2022: 38.4% and 32.9% (Table 1).

The trends in COVID-19 vaccination rates among 5–11-year-old Multiracial children were similar to the overall trend in the county during the study period (Figure 1e,f).

Black 5–11-year-old children consistently had the lowest COVID-19 vaccination rates among all races. Their dose-one vaccination rate was 22.4% on 31 May 2022, 14.3% below the overall vaccination rate (Figure 1e and Table 1); the dose-two rate was 17.2%, 13.4% lower than the overall rate (Figure 1f and Table 1).

### 3.2. Inequalities in Uptake by Children Age-Ethnicity Groups

The trends in COVID-19 vaccination rates among non-Hispanic children closely tracked the overall trends, as people of non-Hispanic ethnicity constitute the majority of the country’s population. Although the vaccination rate trends among Hispanic children were parallel to those among non-Hispanic children, they remained notably and persistently lower (Figure 2).

By 31 May 2022, for example, the dose-one and dose-two vaccination rates among Hispanic 16–17-year-old children were 59.1% and 48%, respectively, versus 67% and 58.9% among their non-Hispanic counterparts (Figure 2a,b and Table 1). At the date, the rates among Hispanic 12–15-year-old children were 55.8% and 45.8%, respectively, versus 64.7% and 57.2% among their non-Hispanic counterparts (Figure 2c,d and Table 1). At ages 5–11 years, the dose-one and dose-two vaccination rates among Hispanic children were 27.6% and 19.1%, respectively, versus 37.8% and 32.0% among non-Hispanic children on 5/31/2-22 (Figure 2e,f and Table 1).

### 3.3. Inequalities in Uptake by Children Age-Race-Sex Groups

The COVID-19 vaccination rate among female children was consistently higher than those of male children regardless of age group, race, and ethnicity (Figure 3 and Figure 4). However, there were major disparities among racial and ethnic groups in the sex differences in COVID-19 vaccination rates (Figure 3 and Figure 4).

#### 3.3.1. 16–17-Year-Olds

The female-male difference (i.e., female minus male) in vaccination rates among 16–17-year-old children rose rapidly from March 2021, when the Moderna vaccine received an EUA from the FDA for the age group, until June 2021 (Figure 3a,b). The difference remained largely stable thereafter. On 31 May 2022, the rates of dose-one and dose-two vaccination among female 16–17-year-old children in the county were 7.3% and 6.6% greater than their male counterparts (Table 1).

The female-male positive difference in the COVID-19 vaccination rates among 16–17-year-old children of Multirace, Some Other Races, and White race, Some Other Races was greater than the overall difference in the county; on the other hand, that among 16–17-year-old children of Black race was smaller than the overall difference (Figure 3a,b). On 31 May 2022, for example, the dose-two vaccination rate among female 16–17-year-old children of Some Other Races was 10.4% greater than that among males in the same age and race groups (Table 1). The difference among Multiracial, White, and Black children was 8.0%, 7.4%, and 5.1%, respectively (Table 1).

An exception to the general trend was 16–17-year-old Asian children: except for the first few months, male Asian children were more vaccinated than female Asian children such that male vaccination rates were 16.9% and 13.8% higher than female rates for dose-one and dose-two vaccine, respectively as of 31 May 2022 (Figure 3a,b and Table 1).

#### 3.3.2. 12–15-Year-Olds

The county-level female-male difference in vaccination rates among 12–15-year-old children increased sharply from April to July 2021 when the vaccination rate in the age group was also sharply increasing (Figure 1c,d and Figure 3c,d). The difference, however, remained largely constant thereafter such that dose-one and dose-two vaccination rates among female 12–15-year-old children were 3.1% and 3.4% greater than male 12–15-year-old children on 31 May 2022 (Figure 3c,d and Table 1). The trends in female-male difference in vaccination rate among White and Black 12–15-year-old children largely paralleled the overall trend, although the difference was consistently higher than average among White children but lower than average among Black children (Figure 3c,d). On 31 May 2022, the dose-one and dose-two vaccination rate among White female children in the age group was 5.0% and 5.2% greater than that among their male counterparts, respectively; 2.6% and 2.5% greater among female Black children of the same age (Figure 3c,d and Table 1).

The female-male difference in vaccination rates was positive until October 2021 but turned negative thereafter, so that dose-one and dose-two vaccination rates among Asian female 12–15-year-old children were 0.1% and 0.9% smaller than their male counterparts on 31 May 2022 (Figure 3c,d and Table 1). The Multirace female-male difference vaccination rates fluctuated around zero during the study (Figure 3c,d). The female-male difference in vaccination rates among 12–15-year-old children of Some Other Race was positive initially but turned negative soon, so the dose-one and dose-two vaccination rates among female 12–15-year-old children of Some Other Race were 3.7% and 0.7% smaller than their male counterparts on of 31 May 2022 (Figure 3c,d and Table 1).

#### 3.3.3. 5–11-Year-Olds

Female 5–11-year-old children were vaccinated more than male children in most racial groups (Figure 3e,f). However, sex differences in the COVID-19 vaccination rate were much smaller among 5–11-year-old children than older children (Figure 3e,f). For example, the dose-two vaccination rate in general and among White and Black female 5–11-year-old children was, respectively, 0.3%, 0.0%, and 0.5% higher than that in their corresponding female counterparts on 31 May 2022 (Figure 3e,f and Table 1).

The highest female-male difference in COVID-19 vaccination rate was measured among 5–11-year-old children of Some Other Races and Multirace (Figure 3e,f). On 31 May 2022, female 5–11-year-old children of Some Other Races were, respectively, 4.7% and 4.9% more dose-one and dose-two vaccinated than their male counterparts; the female-male difference in the vaccination rates among Multiracial 5–11-year-old children were 1.1% and 1.1%, respectively (Figure 3e,f and Table 1).

Opposed to the common trend, female Asian 5–11-year-old children were less vaccinated than their males in the race and age group. On 31 May 2022, for example, dose-one and dose-two vaccination rates among female 5–11-year-old Asian children were, respectively, 0.3% and 2.8% less than that among their male counterparts (Figure 3e,f and Table 1).

### 3.4. Inequalities in Uptake by Children Age-Ethnicity-Sex Groups

COVID-19 vaccination rates among 16–17-year-old female children were persistently greater than male children of both Hispanic and non-Hispanic ethnicities (Figure 4a,b). By 31 May 2022, the dose-one and dose-two COVID-19 vaccination rates among Hispanic 16–17-year-old female children were respectively 8.2% and 7.0% greater than their male counterparts, compared to 7.1% and 6.5% female-male differences among non-Hispanic 16–17-year-old children (Figure 4a,b and Table 1).

COVID-19 vaccination rates among 12–15-year-old non-Hispanic female children were persistently greater than their male counterparts (Figure 4c,d). By 31 May 2022, the female-male difference in dose-one and dose-two COVID-19 vaccination rates among non-Hispanic 12–15-year-old children reached 3.5% and 3.8%, respectively, (Figure 4c,d and Table 1). The female-male difference in vaccination rates among Hispanic 12–15-year-old children gradually shrank and reached –0.6% and 0.1 for doses one and two, respectively, on 31 May 2022 (Figure 4c,d and Table 1).

The female-male difference in COVID-19 vaccination rates was much smaller among 5–11-year-old children of Hispanic and non-Hispanic ethnicities than older children (Figure 4e,f). The dose-one and dose-two female-male-difference in vaccination rates among Hispanic children were 1.7% and 19.6%, respectively, compared to 0.2% and 0.2% difference for non-Hispanic children on 31 May 2022 (Figure 4e,f and Table 1).

The sharp fluctuations in female-male differences for certain racial/ethnic groups (e.g., Asian and Some Other Races) are due to the smaller number of vaccinated children in these categories. For instance, the total numbers of female and male Asian children in the age group 16–17 were 241 and 246, respectively, compared to 4180 and 3803 for 16–17-year-old White children. As a result, even minor variations in the number of vaccinated Asian children over time would lead to a noticeable fluctuation in the trend.

## 4. Discussion

The overall dose-one vaccination rate was higher than the overall dose-two vaccination rate for children of all age groups, and the differences were statistically significant. After the EUA was approved, dose-one COVID-19 vaccination coverage among all age groups experienced a steep increase following a steady increase at a much lower pace. A previous study discovered similar findings: a rapid increase in vaccine uptake during the first 2 months of vaccine availability in the U.S., followed by a plateau [24]. The plateau following the rise might reflect unpleasant experiences; hence, techniques to alleviate the negative dose-one vaccine experiences and associated pain may increase vaccine uptake in children [25]. It could also reflect parents’ opinions on children’s COVID-19 vaccination, as parental COVID-19 vaccination status is one of the most significant predictors of pediatric vaccination status [19,25,26,27,28,29,30], highlighting the need for initiatives aimed at fostering parental trust in the COVID-19 vaccines [28].

The overall rate for 5–11-year-olds was lower than 12–15-year-olds, in turn, lower than 16–17-year-olds, with statistically significant differences (Appendix A). According to the Census Bureau’s analysis of the Household Pulse Survey data, parents of children aged 5–11 years were reluctant to vaccinate their children for concerns related to safety, potential side effects, disbelief in the COVID-19 vaccine, lack of doctor’s recommendation, or unavailability of vaccines [31]. The reasons for parental concern for children around 12–17 years of age centered around side effects following the vaccine dose, mistrust in the government, and the view that the children did not require the vaccine [32]. Moreover, vaccine policy and delivery systems for adolescent vaccines differ from those for childhood vaccines, and sometimes, these disparities may eclipse the effect of parental hesitancy on adolescent vaccine uptake. For instance, vaccine policies for preschool children are structured in a way that numerous health supervision visits are supported during the first 4 years of life, along with legal mandates for childhood vaccines as a prerequisite for school enrollment [33].

Comparing this study’s results to one conducted in New York City, the rate of White children receiving at least one COVID-19 vaccine in Jefferson County was greater than that in New York City by May 2022 [27]. Jefferson County, a less densely populated urban area, might have experienced higher vaccination rates due to effective local vaccination policies, better community engagement, tailored outreach, and accessible vaccination campaigns than New York City [27]. Additionally, compared to the U.S., Jefferson County White children had a higher dose-two COVID-19 vaccination rate [19]. These higher COVID-19 vaccination rates in Jefferson County could suggest effective local policy mandates that might not be strictly enforced nationwide.

The COVID-19 vaccination rate among Jefferson County Asian children was lower than the national average [19]. Nonetheless, their COVID-19 vaccine uptake was one of the highest in the county, except for 12–15 years (Figure 1). There are significant socioeconomic disparities among Asians [34]. For example, foreign-born Asian Americans have lower vaccination rates than their U.S.-born counterparts [35]. Vaccine uptake rates among the county’s Asian children likely reflect that Kentucky ranked as the fourth-leading state in 2023 for the number of refugees arriving compared to other states [36]. Consistent with this study’s findings, a nationwide study reported a notably higher COVID-19 vaccination rate among Asian children aged 5–11 years compared to other racial groups within the same age group [37]. This study’s findings align with similar racial disparities observed in influenza vaccine uptake among children during the 2019–2020 season, which demonstrated significantly lower vaccination rates among other racial groups compared to Asians and non-Hispanic White [37,38]. After controlling for covariates, Asian children aged 5–11 years and 12–17 years had greater odds of receiving COVID-19 vaccination than their non-Hispanic White peers [37].

The COVID-19 vaccination rate among children identifying as Black was consistently the lowest among all racial groups in Jefferson County. This finding aligns with findings from national data, but the Black-White gap in children’s vaccination is wider in Jefferson County [27]. In New York City, for example, there was a much smaller Black-White gap, which was eliminated upon accounting for neighborhood and school time-invariant characteristics [27]―suggesting that neighborhoods’ and schools’ characteristics and resources play a crucial role in alleviating vaccination disparities.

Hispanic children’s COVID-19 vaccination rate was the second lowest among racial and ethnic groups in Jefferson County. The low Hispanic children vaccination rate aligned with the national trend but was lower in Jefferson County than the national average [24]. Also, there was a large and persistent gap in dose-one and dose-two COVID-19 vaccination rates between Jefferson County Hispanic and non-Hispanic children. The gap was notably wider among the 5–11-year-olds. A higher hesitancy among Hispanic parents to immunize their children with the COVID-19 vaccine as opposed to non-Hispanic White parents explains the finding [39]. Recent research on the H1N1 influenza vaccine and seasonal influenza vaccine revealed an intention-action gap among the Hispanic parental group: despite a high intention to vaccinate their children, the actual vaccination rates remained low compared to their non-Hispanic counterparts. The cultural principle of respect among Hispanics, which may discourage asking questions out of fear of being disrespectful to the physicians, could be one of the factors contributing to vaccine hesitancy in the U.S. Hispanic population [40]. Furthermore, under-vaccination among Spanish-speaking Hispanic families might arise due to social or cultural pressures [41].

Most key vaccination rate differences across racial and ethnic groups were statistically significant, particularly at the end of the study period (Appendix A). Economic status can potentially explain the low vaccination rates among Jefferson County Black and Hispanic children, as the poverty rate is the highest among the Black Jefferson County residents (26.5%), followed by the Hispanic residents (24.0%) [42]. Parents with lower incomes often encounter difficulties in taking time off from work to vaccinate their children or to care for them if they experience side effects from the vaccine [43]. Policies that support paid sick leave for parents and caregivers of young children have been associated with higher odds of utilization of medical visits for children, including flu vaccines [44]. Another factor pivotal in parental hesitancy is whether their children are publicly insured, such as through Medicaid, in conjunction with the impact of parents’ lower socioeconomic status [45]. Additional barriers preventing lower-income parents from seeking vaccinations for their children include a lack of pediatric and family medicine practice facilities to cater to care, transportation difficulties, and COVID-19 vaccine-hesitant parents [45,46]. Black-White and Hispanic-non-Hispanic disparities in vaccine uptake are also influenced by structural racism, mistrust, and misinformation, which underscores the need for interventions to mitigate these structural barriers [24]. This study’s findings were further supported by an analysis of influenza vaccination that showed markedly lower rates among non-White children than among White children [47]. Another study revealed that a wide disparity between the household incomes of White and Black children increased over the years, playing a role in detraining their immunization coverage [48]. In addition to income, several factors highlighted disparities in healthcare access, with White children visiting physicians twice as frequently as their minority peers.

Furthermore, physicians who primarily treat African American patients often have less clinical training and limited access to healthcare resources [48]. African American patients often received less timely treatment for various illnesses compared to their White counterparts [49]. These biases, whether unconscious or conscious, held by healthcare providers may influence patient expectations and interactions, potentially discouraging minorities from seeking preventive care services, which could carry on in determining their vaccine uptake [48]. Furthermore, the minority parents who receive these negative experiences who are more concerned with immunization safety are less likely to immunize their children [48]. Previous studies concluded that minority parents are more inclined to believe that children receive more vaccines than required and that their children would contract the illness from the vaccine itself [50,51].

The female-male difference in COVID-19 vaccine uptake was statistically significant in the 12–15 and 16–17 children age groups (Appendix A). While the difference has been shown in other studies [31], analyzing COVID-19 vaccination trends by sex within age-race and age-ethnicity combinations among children is unique. A key finding of this study was that females of most races surpassed males in receiving the COVID-19 vaccine, except for Asian children. Parental perceptions of risk and the importance of vaccination may vary by sex, influencing the likelihood of vaccination for female and male children differently. A similar example is the lower Human Papillomavirus (HPV) vaccination rate among male than female children, as parents of sons perceive a lesser risk of HPV infection [52]. In a study conducted in the Southern US, White women had significantly higher odds than White men in believing that the COVID-19 vaccine was safe [53], which may explain the higher vaccine uptake among White female children compared to White male children in Jefferson County. In a manner akin to this study, national data also revealed that non-Hispanic White females (50.5%) exhibited slightly higher COVID-19 vaccination rates than their male (49.5%) counterparts [54]. In Jefferson County, Black female children across all ages were more likely to receive the COVID-19 vaccination when compared to their male counterparts. Cunningham-Erves, J. concluded that in the Southern U.S., Black men had higher odds of expressing concerns about the potential side effects of the COVID-19 vaccine than Black women [53]. While examining age-ethnicity-sex, by the end of May 2022, this study showed both Hispanic and non-Hispanic girls in all age groups had notably higher COVID-19 vaccination rates than boys of the same age group in Jefferson County, except for the age group 12–17 years. This pattern aligns with the findings of another study, where non-Hispanic females surpass the non-Hispanic males in their COVID-19 vaccination rates [54].

### Limitations

This study’s findings are constrained by data limitations. For example, if a piece of demographic information was not reported when receiving a dose, it was imputed using the most commonly reported information across other doses. As such, the finding could have been influenced by data entry errors.

Another major challenge in this data was inconsistent reporting of Multiracial race and Some Other Races across vaccine doses, resulting in the overrepresentation of these racial groups in the vaccination data compared to their county population according to the U.S. censuses. As there is no definitive method to address these inconsistencies in race reporting, an algorithm was developed [20]. Moreover, an unknown ethnicity was labeled as non-Hispanic to adjust the number of vaccinated individuals by ethnicity to their corresponding population in the county. The practice might have resulted in an underestimation of the Hispanic population.

The overrepresentation of some racial groups in the immunization registries relative to population statistics and variable race assignments across various encounters may provide similar difficulties for future studies. Therefore, improving the algorithm to tackle the challenges can improve the quality of socio-epidemiological studies and potentially public health policymaking.

## 5. Conclusions

Comprehensive, multidimensional demographic analyses of COVID-19 vaccine uptake among children are essential to understanding gaps in children’s vaccination in greater detail and to devise effective COVID-19 vaccination promotion strategies that are age, sex, race, and ethnicity-appropriate. Such analyses need to be done locally, as the extent of demographic inequalities in vaccination can substantially vary geographically. However, local vaccine uptake can be impacted by national policy to address demographic inequality observed at the local level. For example, health insurance may play a role in perceived and actual ease or difficulty in accessing vaccinations, however, decisions about public safety nets or insurance coverage are not made at a local level. Therefore, national and state policies impact city/county geographic patterns.

Additionally, community-engaged vaccine campaigns developed by public health in conjunction with partnerships can contribute to more successful vaccination rates among marginalized populations. Experiences from COVID-19 vaccination campaigns showed that successful localized strategies to reach high-risk populations were built upon responding to the populations’ unique needs, integrating private partnerships, engaging trusting messengers and spokespersons, and devising culturally responsive messaging. Also, mobilizing public health resources and effective planning and coordination are critical to overcoming resource shortfalls and logistic challenges in the face of high infection rates but limited vaccine supply.

## Figures and Tables

**Figure 1 healthcare-13-01019-f001:**
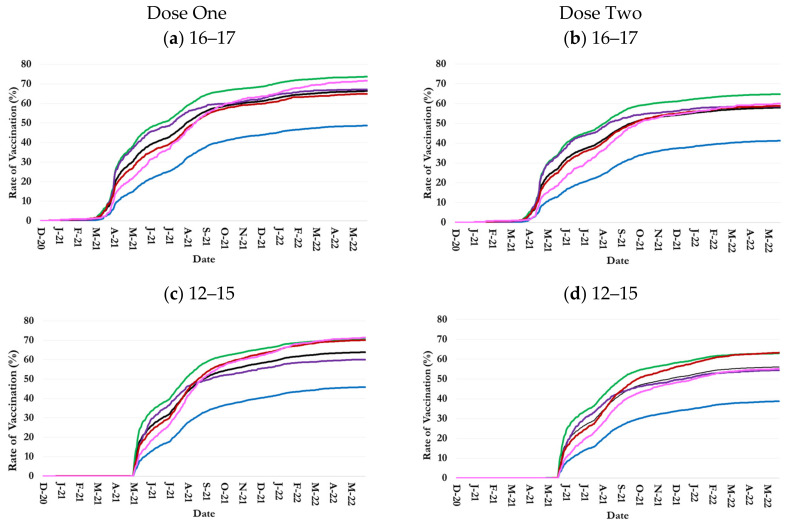
Trends in one- and two-dose COVID-19 vaccination rates by age and race in Jefferson County, Kentucky.

**Figure 2 healthcare-13-01019-f002:**
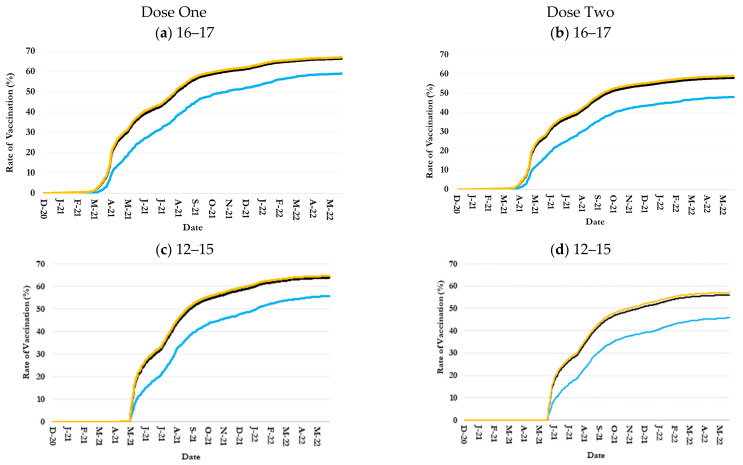
Trends in one and two-dose COVID-19 vaccination rates by age and ethnicity in Jefferson County, Kentucky.

**Figure 3 healthcare-13-01019-f003:**
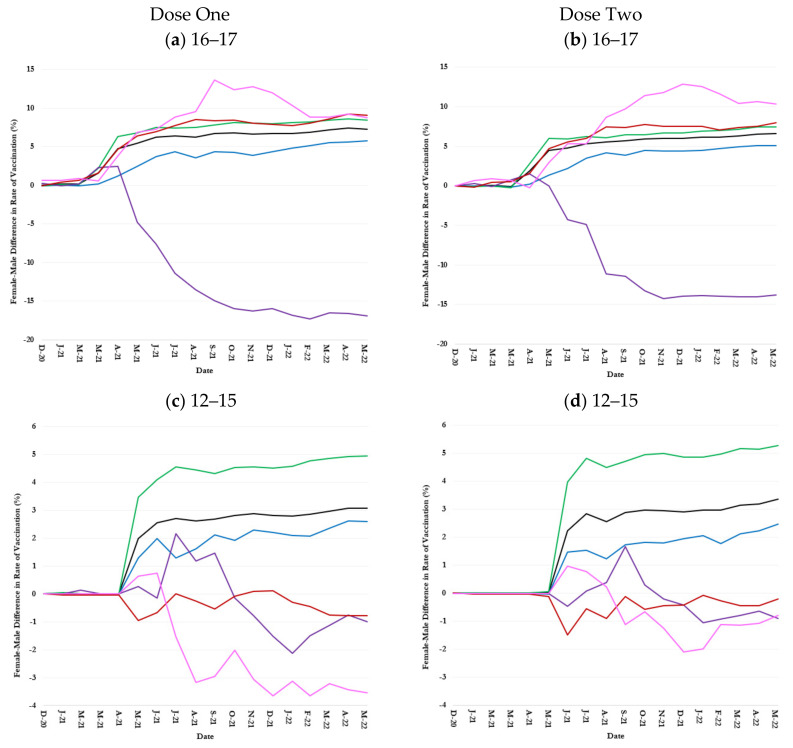
Sex difference (female minus male) in one and two-dose COVID-19 vaccination rates over time by age and race in Jefferson County, Kentucky.

**Figure 4 healthcare-13-01019-f004:**
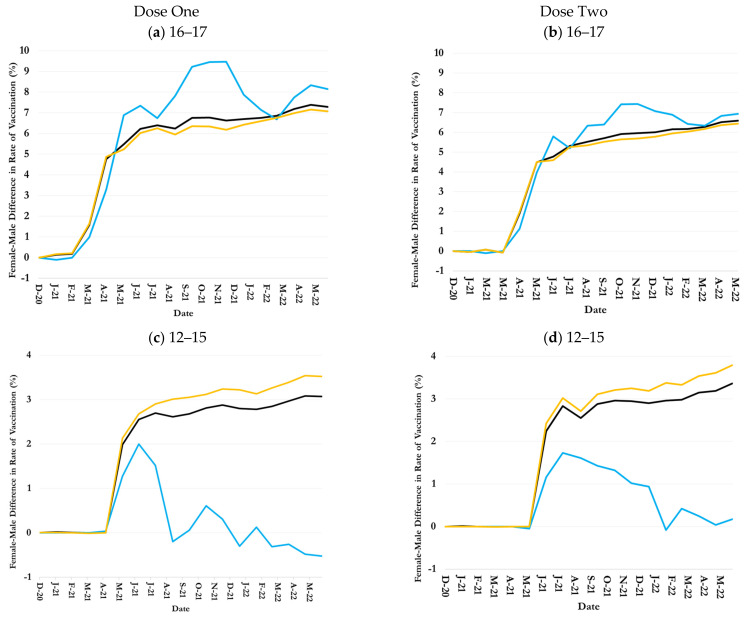
Sex difference in one- and two-dose COVID-19 vaccination rates over time by age and ethnicity in Jefferson County, Kentucky.

**Table 1 healthcare-13-01019-t001:** Vaccination rate differences by children age-race-sex and age-ethnicity-sex combinations on 31 May 2022 in Jefferson County, Kentucky.

Dose Number		Race						Ethnicity	
		White	Black	Asian	Some Other Races	Multi-Racial		Hispanic	Non- Hispanic
(a) 16–17									
	Vaccination Rate (%)								
1	66.4	73.7	48.7	67.2	71.7	64.8		59.1	67.0
2	58.0	64.8	41.2	58.6	60.0	58.8		48.0	58.9
	Difference from the Overall Vaccination Rate (%)								
1	0.0	+7.3	–17.7	+0.8	+5.3	–1.6		–7.3	+0.6
2	0.0	+6.8	–16.8	+0.6	+2.0	+0.8		–10.0	+0.9
	Female Vaccination Rate (%)								
1	70.0	77.9	51.7	62.0	77.5	69.4		63.0	70.5
2	61.3	68.5	43.8	54.5	66.3	62.8		51.4	62.2
	Female Minus Male Difference in Vaccination Rate (%)								
1	+7.3	+8.5	+5.8	–16.9	+8.8	+9.1		+8.2	+7.1
2	+6.6	+7.4	+5.1	–13.8	+10.4	+8.0		+6.9	+6.4
(b) 12–15									
	Vaccination Rate (%)								
1	63.9	70.5	45.9	60.1	71.5	70.1		55.8	64.7
2	56.1	62.9	38.8	54.4	55.2	63.3		45.8	57.2
	Difference from the Overall Vaccination Rate (%)								
1	0.0	+6.6	–18.0	–3.8	+7.6	+6.2		–8.1	+0.8
2	0.0	+6.8	–17.3	–1.7	–0.9	+7.2		–10.3	+1.1
	Female Vaccination Rate (%)								
1	65.5	73.1	47.2	59.9	69.7	69.7		55.4	66.5
2	57.8	65.6	40.0	54.3	54.9	63.2		45.8	59.1
	Female Minus Male Difference in Vaccination Rate (%)								
1	+3.1	+5.0	+2.6	–0.1	–3.7	–0.8		–0.6	+3.5
2	+3.4	+5.3	+2.5	–0.9	–0.7	–0.3		+0.1	+3.8
(c) 5–11									
	Vaccination Rate (%)								
1	36.7	43.4	22.4	43.4	50.9	38.4		27.6	37.8
2	30.6	35.5	17.2	36.7	37.0	32.9		19.1	32.0
	Difference from the Overall Vaccination Rate (%)								
1	0.0	+6.7	–14.3	+6.7	+14.2	+1.7		–9.1	+1.2
2	0.0	+5.6	–13.4	+6.1	+6.0	+2.9		–11.5	+1.4
	Female (%)								
1	36.9	41.4	22.5	43.3	53.2	38.9		28.5	37.9
2	30.8	35.6	17.4	35.8	39.4	33.5		19.6	32.1
	Female Minus Male Difference in Vaccination Rate (%)								
1	+0.4	+0.2	+0.2	–0.3	+4.7	+1.1		+1.8	+0.2
2	+0.3	+0.0	+0.5	–2.8	+4.9	+1.1		+0.8	+0.2

## Data Availability

The datasets analyzed in this cannot be publicly accessible.

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
