# Peer review of "COVID-19 Vaccine Uptake Inequality Among Children: A Multidimensional Demographic Analysis"

_healthcare, 2025, doi:10.3390/healthcare13091019_

Round 1
Reviewer 1 Report
Comments and Suggestions for Authors
Please check the attachment

Reviewer 2 Report
Comments and Suggestions for Authors
You did mention that income may have an impact on vaccination rates. Was parental education mentioned?
I think a paragraph covering Jefferson County, KY would be crucial. What is the gender breakdown, race/ethnicity, overall population, income, rural/urban, suburban of the county? It helps the reader being to relate your results to the overall population.
Why was there a lack of statistical calculations? Can we surmise that there are statistically significant differences between the racial groups by vaccination? How about age? Dose #1 vs. Dose #2 for gender and race?. Descriptive statistics do not make the comparisons to pick out significant differences. This affects any claim that are made by differences in vaccination rates. The differences may appear either significant or insignificant but may be the opposite when actually tested.
Round 2
Reviewer 2 Report
Comments and Suggestions for Authors
Thank you in answering or commenting on most of my recommended changes/edits.